# Protocol for the COG-UK hospital-onset COVID-19 infection (HOCI) multicentre interventional clinical study: evaluating the efficacy of rapid genome sequencing of SARS-CoV-2 in limiting the spread of COVID-19 in UK NHS hospitals

James Blackstone [1], Oliver Stirrup,[2] Fiona Mapp [2], Monica Panca,[1] Andrew Copas [2], Paul Flowers [3], Leanne Hockey [1], James Price,[4] David Partridge [5], Christine Peters,[6] Thushan de Silva [7], Gaia Nebbia [8], Luke B Snell [8], Rachel McComish,[1] The COVID-19 Genomics UK (COG-UK) Consortium, Judith Breuer[9]

For numbered affiliations see end of article.

**Correspondence to**
Professor Judith Breuer;
j.breuer@ucl.ac.uk

## ABSTRACT

**Objectives** Nosocomial transmission of SARS-CoV-2 has been a significant cause of mortality in National Health Service (NHS) hospitals during the COVID-19 pandemic. The COG-UK Consortium Hospital-Onset COVID-19 Infections (COG-UK HOCI) study aims to evaluate whether the use of rapid whole-genome sequencing of SARS-CoV-2, supported by a novel probabilistic reporting methodology, can inform infection prevention and control (IPC) practice within NHS hospital settings.

**Design** Multicentre, prospective, interventional, superiority study.

**Setting** 14 participating NHS hospitals over winter–spring 2020/2021 in the UK.

**Participants** Eligible patients must be admitted to hospital with first-confirmed SARS-CoV-2 PCR-positive test result >48 hour from time of admission, where COVID-19 diagnosis not suspected on admission. The projected sample size is 2380 patients.

**Intervention** The intervention is the return of a sequence report, within 48 hours in one phase (rapid local lab processing) and within 5–10 days in a second phase (mimicking central lab), comparing the viral genome from an eligible study participant with others within and outside the hospital site.

**Primary and secondary outcome measures** The primary outcomes are incidence of Public Health England (PHE)/IPC-defined SARS-CoV-2 hospital-acquired infection during the baseline and two interventional phases, and proportion of hospital-onset cases with genomic evidence of transmission linkage following implementation of the intervention where such linkage was not suspected by initial IPC investigation. Secondary outcomes include incidence of hospital outbreaks, with and without sequencing data; actual and desirable changes to IPC actions; periods of healthcare worker (HCW) absence. Health economic analysis will be conducted to determine cost benefit of the intervention. A process evaluation

## Strengths and limitations of this study

► Harnesses infrastructure of UK's existing national COVID-19 genome sequencing platform.
► First prospective interventional study to assess effectiveness of genomic sequencing for infection prevention and control in an unbiased patient selection in secondary care.
► Awarded UK National Institute of Health Research Urgent Public Health status, ensuring prioritisation of recruitment.
► A limitation is that the study does not have a randomised controlled design.

using qualitative interviews with HCWs will be conducted alongside the study.

**Trial registration number** ISRCTN50212645. Pre-results stage. This manuscript is based on protocol V.6.0. 2 September 2021.

## INTRODUCTION
### Background

Hospitals are recognised to be a major risk for the spread of infections despite the universal introduction of infection control measures. For SARS-CoV-2, nosocomial spread of infection presents an additional and significant health risk to patients and healthcare workers (HCW).[1] During epidemics, infection prevention and control (IPC) practice is further complicated by the difficulties of distinguishing community and hospital-acquired infections. This can lead to erroneous identification of nosocomial

transmission, leading to unnecessary IPC efforts. True nosocomial transmission events may be missed with appropriate interventions not performed, thereby putting patients and HCW at increased risk. The epidemiological determination of infection timing for SARS-CoV-2 is made especially challenging by its prolonged incubation period in distinguishing community from nosocomial transmission.

There is now good evidence that genome sequencing of epidemic viruses, together with standard IPC, better defines nosocomial transmissions and, depending on the virus, better identifies routes of transmission, than IPC alone.[2–4] The development of rapid sequencing methods enables sequencing of potentially linked or unlinked SARS-CoV-2 genomes within 48 hours. This timescale is short enough to inform clinically relevant IPC decisions in near-real time. Although some studies have described the prospective use of viral sequencing to inform infection control for SARS-CoV-2, none has prospectively evaluated the impact of sequencing on the incidence of nosocomial infection or on infection control actions across all cases with hospital onset.[5–7]

While SARS-CoV-2 has a low mutation rate (estimated at around 2.5 changes per genome per month), sufficient viral diversity exists to identify cases where patient and HCW infections that are clustered in time and space are in fact due to different SARS-CoV-2 genotypes.[8] Such information could rapidly exclude nosocomial transmission as the cause of the cluster and redirect IPC intervention to where needed most.

Confirmation of SARS-CoV-2 transmission to patients and HCWs may be more challenging with a single-observed mutation between two genomes, feasibly representing anything between 1 and 10 transmissions. Identical genomes will not necessarily evidence a close-link between two cases. Nonetheless, by comparing genotypes detected within the hospital setting and the surrounding community, it may be possible to reveal unsuspected nosocomial transmission where comparatively uncommon genotypes are apparently linked or cluster in time and space.

The Consortium Hospital-Onset COVID-19 Infections-UK (COG-UK) initiative aims to sequence as many SARS-CoV-2 viruses as possible across the UK for public health planning. It also provides an important and unique opportunity to test whether viral sequence data produced in near-real-time could also reduce uncertainties around nosocomial transmission events, better direct IPC effort, improve hospital functioning and reduce the role of hospitals as a source of infection to the community.[9]

COG-UK HOCI* will harness the COG-UK sequencing platform, with its mixed model of smaller sequencing hubs located close to hospitals and a large centralised hub sequencing most viruses. It will identify not only whether rapid viral sequencing is useful for patient management, but how time-critical this might be; turnaround times for sequence data from a central hub are likely to be longer (5–10 days) than those from local sequencing hubs (<48 hours).

*Note that while HOCI was the preferred term at the study's inception, evolution of the terminology now favours references be made to 'hospital-onset SARS-CoV-2 infections'

## Objective

The study will evaluate the contribution of whole-genome sequencing combined with a novel viral sequence report design to IPC investigation and response to cases of hospital-onset COVID-19 infection, and whether this can reduce the overall incidence of hospital-acquired infections.[10]

## METHODS AND ANALYSIS
### Study design

COG-UK HOCI is a prospective, interventional, superiority clinical study, comprising three distinct phases with a possible fourth, dependant on interim data analysis.

In the first phase, all sites will collect baseline (non-interventional) eligible patient data for a period of 4 weeks to characterise each site's usual practice in infection control in response to hospital-onset COVID-19 cases. This phase may include standard of care use of genome sequencing (eg, limited outbreak response analysis).

In the second and third phases, the study intervention will be applied on top of standard of care infection control practices.

The second phase requires 'rapid' turnaround of genome sequencing and sequence linkage report generation (ie, within 48 hours of first diagnostic SARS-CoV-2 PCR-positive result). This phase will be applied to all hospital onset COVID-19 cases meeting the eligibility criteria over an 8-week period.

The third phase is similar to the second, except that a 5–10 day turnaround time of genome sequencing and sequence report generation should be applied to mimic the use of a central sequencing laboratory. This phase will apply to all sites and last for 4 weeks.

The second and third phases may be applied in the reverse order at some sites, both for logistical reasons (ie, fine-tuning of rapid turnaround of whole-genome sequences) and also to ensure differences between sites in the calendar dates of each phase.

On review of interim analysis data, the study's joint oversight committee may recommend a fourth phase for all sites comprising a second baseline period; this would be applied where the initial baseline data collection period occurred at a time of very high or low COVID-19 prevalence at the sites, whereby collecting data on standard practice could be unviable.

This will be a sequential study, with each National Health Service (NHS) Trust acting as its own control.

The total study duration per site will accordingly be 16–20 weeks, though it is likely that pauses in data collection will occur over the winter holiday break period due to most sequencing labs closing or moving to skeletal operations during this time.

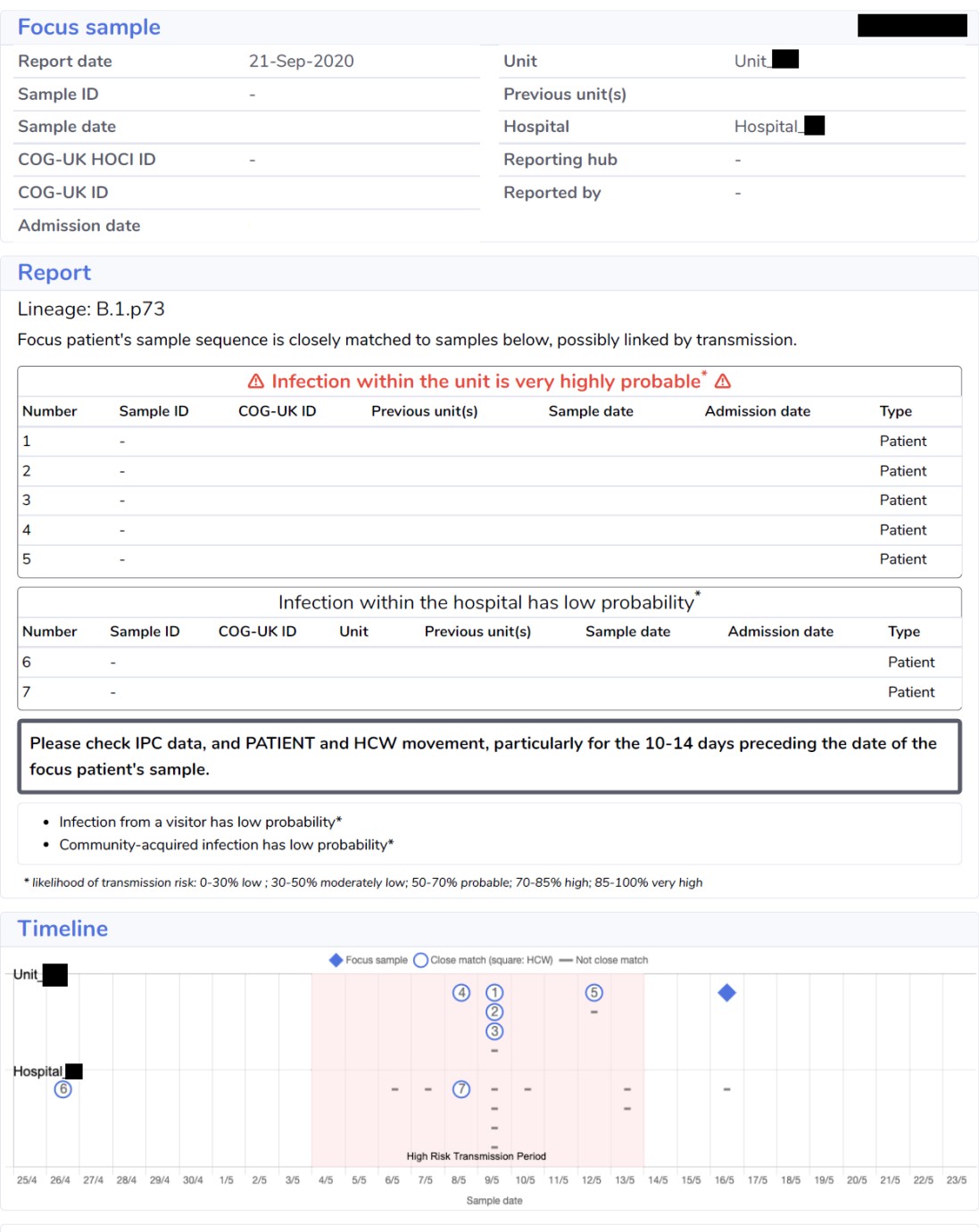

**Figure 1** Example of HOCI Sequence Reporting Tool (SRT) Report Output. COG-UK HOCI, COG-UK Consortium Hospital-Onset COVID-19 Infections; HCW, healthcare worker; IPC, infection prevention and control.

## Intervention

### Overview

The study intervention is a SARS-CoV-2 genomic sequencing data report (see figure 1) delivered to the NHS site's IPC teams, either within 24–48 hours of the sample from the patient being confirmed as positive for SARS-CoV-2 (rapid genomic sequencing locally) or within 5–10 days (local genomic sequencing to mimic use of a centralised lab).[11]

Microbiology and IPC teams will be trained to interpret the results. An expert sequence interpretation team (a subset of the Study Team) will be available 7 days a week by phone and online to discuss results where required with IPC teams and to provide guidance on best practice.

### Genomic sequence reporting tool

The genomic sequence report tool combines epidemiological and consensus sequence data in order to

provide a rapid assessment of the probability of hospital acquired infection (HAI) among new HOCI study cases and to identify infections that could plausibly constitute a hospital outbreak event.[12]

The internal calculations use a combination of admission-to-symptom-onset intervals and differences between the observed proportion of close sequence matches (defined as a maximum pairwise difference of two-single nucleotide polymorphisms; SNPs) for viral samples obtained from various locations (ie, same ward, same hospital, within the community) to estimate the probability that the patient's SARS-CoV-2 infection was acquired in hospital.

The report generation algorithm is designed to run quickly and reliably, without the need for local model checking, thereby reducing the need for expert bioinformatics input during operation.

### Sequence reports

The summary report for each focus sequence submitted, corresponding to a single HOCI case, will comprise:

1. The lineage assignment for the focus sequence.
2. A list of the details of any close sequence matches from samples on the same ward as the focus sequence in the previous 3 weeks, with estimated probability of infection having occurred from a source on the ward (reported as low, moderately low, probable, high, very high).
3. A list of the details of any close sequence matches from samples obtained within the hospital but not same ward as the focus sequence in the previous 3 weeks, with estimated probability of infection having occurred from a source in the institution (reported as low, moderately low, probable, high, very high).
4. Estimated probability of infection from a visitor to the ward (reported as low, moderately low, probable, high, very high).
5. Estimated probability of community-acquired infection (reported as low, moderately low, probable, high, very high).
6. A graphical summary displaying sample dates of close sequence matches at the ward and hospital levels, along with the total number of samples obtained over the previous 3 weeks.

A detailed report will also be returned to virology labs for each focus sequence, containing additional details of all the recent sequences obtained within the given ward and hospital that have contributed to the output summarised in the summary report, and their similarity with respect to the focus sequence.

### Allocation of intervention

All sites will engage in the various study phases sequentially; there will be no allocation of intervention either by site or at patient level for this study.

### Population

#### Setting

Fourteen NHS Trusts/Heath Boards across England and Scotland will participate. Sites will be set-up either as all hospitals within a Trust or a single hospital selected from within the Trust. This decision will be site-led and based on available research team, infection control team and sequencing resource. Sites will be selected to span tertiary referral centres through to district hospitals, primarily in urban or suburban settings. Screening and routine testing of patients and HCWs will follow local health guidance.

### Inclusion criteria

Patients will be considered eligible only where they are an inpatient with first confirmed positive test for SARS-CoV-2 >48 hours after admission, where they were not suspected to have COVID-19 at time of admission.

Participants may be of any age to be included in the study.

There are no exclusion criteria.

### Recruitment

Viral sequencing will be attempted for every confirmed case of SARS-CoV-2 in hospital patients and HCW, but it is not possible to assess clinical and infection control outcomes for every confirmed case. This study will, therefore, focus on the subset of patients with hospital-onset SARS-CoV-2, since this is where the additional knowledge potentially provided by viral sequencing is likely to have the greatest impact for IPC teams. HCWs will not be enrolled as index cases but will be part of the reference sequence data set where available. Patient samples will be collected and sequenced per standard NHS Trust practice in support of pre-existing site arrangements to provide SARS-CoV-2 sequence data to COG-UK in support of national genomic surveillance for public health.

### Patient consent

Consent for participant (both patient and HCW) involvement will not be sought for COG-UK HOCI study. This approach relies on the Health Service (Control of Patient Information) Regulations 2002 (SI 1438), specifically Regulation 3 (Communicable disease and other risks to public health), and Regulation 7 (the processing of confidential information for medical research).

### Study outcomes

#### Primary outcomes

1. Incidence rate of PHE/IPC-defined SARS-CoV-2 HAIs (defined as SARS-CoV-2 cases with an interval of ≥8 days from admission to symptom onset, if known, or sample date), measured as incidence rate of recorded cases per week per 100 inpatients, during each phase of the study.
2. Identification of linkage to individuals within an outbreak of SARS-CoV-2 nosocomial transmission using sequencing report data for HOCIs in whom this was not identified by presequencing IPC evaluation, for each enrolled patient during study phases in which the sequence reporting tool is in use.

## Secondary outcomes

1. Incidence rate of IPC-defined SARS-CoV-2 hospital outbreaks, defined as cases of hospital transmission linked by location and with intervals between diagnoses no greater than 28 days, measured as incidence rate of outbreak events per week per 100 inpatients during each phase of the study.
2. Incidence rate of IPC+sequencing-defined SARS-CoV-2 hospital outbreaks involving HOCI cases, defined as for IPC-defined SARS-CoV-2 hospital outbreaks with the additional condition of clustering of viral sequences and measured as outbreak events per week per 100 inpatients during study phases in which the sequence reporting tool is in use. Genetic clusters are defined as having maximum viral sequence pairwise SNP distance of 2 between each individual included and their nearest neighbour within the cluster.
3. Changes to IPC actions implemented following receipt of SARS-CoV-2 sequence report, for each enrolled patient during study phases, in which the sequence reporting tool is in use.
4. Changes to IPC actions that would ideally have been implemented but may not have been following receipt of SARS-CoV-2 sequence report, for each enrolled patient during study phases in which the sequence reporting tool is in use.
5. Health economic benefit of both slow and rapid sequencing reports to IPC against baseline.
6. The number of HCW periods of sickness/self-isolation, assessed as a proportion of the number of staff usually on those wards impacted by HOCI cases, for all phases of the study.

In collaboration with IPC teams at NHS sites, the outcome data above will be collected using case report forms (CRFs) specifically designed for this study. These will be prospectively completed during the course of the study and stored in a central study database.

## Exploratory outcomes

Additionally, descriptive summaries of sequence report results will be generated, including number of close sequence matches on ward and within hospital; probability of infection source; whether HCWs are reported within close sequence matches.

For the process evaluation, the qualitative team will seek to understand how the intervention worked in practice across a representative sample of study sites (n=5). This will include how the context shaped the intervention; how key intervention components and causal mechanisms operated for IPC teams and hospital planners, and how the intervention changed the study outcomes.

## Sample size and power

The projected total sample size is 2380 patients.

There is uncertainty in the number of HOCIs that will be identified at each site during each of the intervention periods, with the rapid testing phase being 8 weeks' duration. Based on clinical experience of first wave and discussion with the principal investigators, we assume there may be an average of 10 HOCIs/week per site during this intervention period, a total of 80 per site. Within a typical site this will allow us to estimate the proportion of HOCIs with genotypic linkage to any other case(s) not detected by IPC processes with minimum precision of±9.4%. Similarly, we can estimate the proportion of HOCIs where an action is taken that would not have occurred without sequencing within±9.4%. We shall also calculate pooled estimates of these proportions across the 14 sites, leading to estimation within±6.5% assuming an intracluster correlation coefficient of 0.05.

Comparing the proportion of HOCIs with genotypic linkage to any other case(s) not detected by IPC processes between rapid testing and slow testing phases across all sites, the study would have at least 80% power to detect a percentage point difference of 11%. This corresponds to a two-sided test with alpha=0.05, considering proportions of 55.5% vs 44.5% which would be associated with minimum power for a difference of this magnitude.

For the outcome of weekly incidence of IPC-defined HOCIs, using an approximate normal distribution for weekly counts there is 86.7% power to demonstrate a reduction from 12 IPC-defined HOCIs per week in the baseline phase to 10 per week during the rapid testing phase across all sites, under 5% significance level two-tailed testing. However, these calculations correspond to a variance of 12 for weekly counts based on the Poisson distribution, but the presence of over-dispersion of weekly counts would lead to a lower power to detect a difference. Using an overdispersion parameter of 0.82 based on retrospective analysis of data from Sheffield and Glasgow (dataset as described by Stirrup *et al*) results in 81% power to detect a reduction in mean weekly incidence from 12.5 to 10.[10]

## Data management and protection

All study documentation at site will be held in restricted access areas and stored securely by study team members. Data will be entered by sites into a secure, validated online database (Elsevier MACRO v4) and accessible only by delegated team members of that site, and by delegated staff from the coordinating centre.

CRFs for the study will identify patients using a unique five-digit study identifier, year of birth and initials. Under the Data Protection Act 2018, the latter identifiers will be considered 'personally identifying' and will be treated as such by both the site team and coordinating centre team.

Where written communication (eg, data queries) on individual patient cases is necessitated between sites and the coordinating centre, only the study identifier should be used in the first instance.

Any transfer of documentation containing personally identifying data between site and coordinating centre will be subject to AES-256 industry-standard encryption.

## Statistical analysis

Summary statistics will be presented for each study phase (baseline, rapid local lab and central lab interventions) and each site, which will be percentages for binary outcomes such as whether transmission linkage for each HOCI was previously undetected. The frequency of IPC-defined HAIs, IPC-defined outbreak events and IPC with sequencing-defined outbreak events will be expressed as rate per week per 100 inpatients.

The outcomes of genotypic identification of transmission linkage not suspected at initial IPC investigation and impact of sequencing on IPC actions are only defined for the intervention periods. For such outcomes, the focus of analysis is to calculate summary statistics overall and for each site, which can be informally compared with the degree to which it is thought each site was able to fully implement the intervention. Variation over time within each site will also be explored, and the proportions will be compared between the rapid sequencing and delayed sequencing intervention periods.

The main analyses for the primary and secondary outcomes will be carried out on an intention-to-treat basis according to the defined study phases. However, sensitivity analyses will be conducted excluding study sites and/or periods with suboptimal implementation of the study intervention, both in terms of overall population sequencing coverage for HOCIs and the turnaround time for sequence reports being returned to IPC teams.

For outcomes defined in both the baseline and intervention periods, such as incidence of IPC-defined HAIs and the number IPC-defined hospital outbreaks, this can be informally compared between the baseline, intervention and (where implementation is justified) final control periods within sites. A formal analysis will be conducted based on negative binomial regression to detect the change in the incidence rate of each event type between baseline, intervention and control phases within site, including the current proportion of inpatients who are SARS-CoV-2 positive as a fixed effect and exposure 'determined' by the number of SARS-CoV-2-negative inpatients in that week. These analyses will also include adjustment for the proportion of HCWs at each site who have received at least one vaccine dose and a smoothed adjustment for calendar time. This will lead to an adjusted incidence rate ratio for the intervention effect, presented with a 95% CI.

Missing data will be identified, and efforts made to obtain the data. In the event that some sites are unable to implement the intervention fully during the intervention period then analysis will be repeated excluding such sites to provide a 'per protocol' analysis.

A detailed statistical analysis plan will be produced prior to commencement of analysis and agreed by the joint Trial Steering Committee and Data Monitoring Committee (TSC-DMC). All statistical tests will use a two-sided p value of 0.05, unless otherwise specified. All statistical analyses will be performed using Stata (StataCorp, College Station, Texas).

## Health economic analysis

We will examine whether rapid SARS-CoV-2 genomic sequencing might lead to measurable economic advantages. A cost-benefit analysis will be conducted looking at the incremental cost or savings for the two sequencing approaches against baseline in the group of sites influenced by the time to sequence data result.

The cost of SARS-CoV-2 genome sequencing, generating the report and additional resources involved in teams training and review of the report, will be obtained from the participating laboratories and study sites.

HOCI resource use will be obtained from hospital records and CRFs supplemented with information obtained from members of the IPC teams at each site to inform IPC action-related cost. Costs will be evaluated from the NHS setting perspective over the study period. Economic benefits include the attributable cost savings from reducing the delay to initiate IPC measures to avert infection transmission, an estimate of the hospital cost savings due to excess bed days and days off work by HCWs.

Mean costs and SD for all phases of the study will be calculated. We will estimate the incremental mean difference in total costs between intervention phases and baseline of the study and 95% CIs.

Deterministic sensitivity analysis will be performed to assess the impact of varying resource use and other relevant parameters to identify variables with the highest impact on costs.

Adjustments will be made for variation in HOCI levels due to impact of B.1.1.7 variant of SARS-CoV-2 in the UK, as well as the national COVID-19 vaccine rollout.

A Health Economic Analysis Plan will be prepared for the study prior to commencing data analysis and will be approved by the TSC-DMC.

## Process evaluation

Process evaluations are now considered integral to understanding the factors which shape outcomes achieved within a study, enrich interpretation of findings and facilitate better understandings of how the intervention may be used in other settings to create sustainable health change.

The process evaluation embedded within the HOCI study aims to understand how the rapid genome-sequencing intervention works in practice across different sites.

The team will first develop initial programme theory for the SRT in advance of implementation. Programme theory describes the salient parts of the context in which the intervention will be implemented, the specific nature of the problem being addressed, the content or components of the intervention, the mechanisms through which the intervention works and how the intervention led to expected and unanticipated outcomes. Including the development of the programme theory as part of the study ensures that the team has derisked the intervention as far as possible by anticipating and mitigating potential problems or limiting factors.[12]

Data will then be gathered using a topic guide based on the programme theory. A purposive sample of HCWs involved in the chain of activity associated with implementing the SRT across five study sites will be interviewed. Interviews will take place during or soon after the rapid phase and focus on how the SRT and HOCI study more broadly have been implemented. A structured thematic analysis of the data will be conducted using the core elements of the initial programme theory, which will then be refined and used to share learning on HOCI and facilitate transferable knowledge and sustainable future healthcare.

### Study timelines

Sites will be opened using a staggered approach from October 2020 to January 2021 in order to provide the greatest likelihood of each phase of site activity covering peaks, troughs and moderate incidences of community prevalence and, therefore, likely hospital admissions of patients with COVID-19.

Patient recruitment at sites will run for 6 months from late October 2020 to end April 2021.

See table 1 for study schedule.

### DISCUSSION

By defining and reporting SARS-CoV-2 genotype frequencies within its sites and comparing to those in the wider community, the study has the potential to overcome some of the inherent barriers to identifying the likely transmission chains. The data generated will provide an accurate as possible a picture, given the constraints of viral genetic diversity, of the number and location of SARS-CoV-2 infections acquired by nosocomial transmission and to an extent inform how these transmissions are occurring.

| Table 1 Study schedule | | | | | | |
|---|---|---|---|---|---|---|
| **Timepoint (site dependent)** | **6 months** | **4 weeks** | **8 weeks** | **4 weeks** | **4 weeks** | **6 months** |
| **Study stage (site dependent)** | **Set up** | **First baseline/ control (daily)** | **Intervention sequencing result <48 hours (daily)\*** | **Intervention sequencing result >4 days (daily)\*** | **Second baseline/ control (where justified, daily)** | **Data cleaning, analysis and reporting** |
| Site identification | X | | | | | |
| Site team discussion on sampling ability, staffing availability and logistics | X | | | | | |
| Initiation of contracts, R&D approvals | X | | | | | |
| NHS samples begin to be processed locally under COG-UK approvals (not study-related) | X | | | | | |
| NHS samples to be processed under COG-UK approvals, either locally or at Sanger | | X | X | X | X | |
| Intervention reports generated | | | X | X | | |
| Intervention reports returned to site ICTs (<48 hour) | | | X | | | |
| Intervention reports returned to site ICTs (>4 days) | | | | X | | |
| ICTs evaluate reports, seeking Expert Sequence Interpretation Team views if needed | | | X | X | | |
| Case reports for HOCIs | | X | X | X | X | |
| Process evaluation—qualitative interviews /analysis | | | X | X | X | x |
| Process evaluation—programme theory development and refinement | x | | x | x | x | x |
| Interim analysis and views from TSC-DMC whether second baseline/control state acceptable | | | | | X† | |
| Data cleaning | | X | X | X | X | X |
| Final data lock and analysis | | | | | | X |
| Reporting/publication | | | | | | X |

\*The order of 'rapid' phase (<48 hour turnaround time) and 'slow' phase may be swapped prior to commencement of either on agreement with the Sponsor. The option is offered to facilitate logistics/set-up at sites.
†TSC-DMC review should take place to determine whether it would be considered ethical to request sites have a second period of baseline/control (where sequencing data is not provided to IPC teams). This would only be on the basis that it is unclear from the initial baseline and intervention comparison whether there is a significant benefit; in cases where it is clear there is either benefit to the intervention or no benefit, then the second baseline would not take place.
COG-UK HOCI, COG-UK Consortium Hospital-Onset COVID-19 Infections; ICT, Infection Control Team; IPC, infection prevention and control; NHS, National Health Service; TSC-DMC, Trial Steering Committee and Data Monitoring Committee.

While COG-UK will provide data on the utility of viral genomics for national public health planning, COG-UK HOCI will quantify the utility of the same data for local management of nosocomial infection, including whether observed benefits are time dependent and deliver the best estimates of how viral sequence data can be used to identify HAIs among HOCIs.

Study outputs will further inform decisions about the likely future use of viral genome sequencing for the management of epidemics and pandemics and how it might best be organised—centralised or diversified—to deliver maximal impact.

### Study monitoring

An independent joint TSC-DMC will be formally responsible for the oversight of the study and ensuring it is conducted in compliance with ICH Good Clinical Practice and other relevant regulations. The TSC-DMC will also advise on the need for the fourth phase of the study (a second baseline period).

The Trial Management Group will be responsible for the execution of the study. Site monitoring will be undertaken by the Trial Manager, based at the UCL Comprehensive Clinical Trials Unit.

Site teams will only report adverse events, which meet both the 'seriousness' threshold and are also considered 'related' to the study intervention. This was considered risk appropriate for the study as no patient-specific procedures are undertaken and has been approved by the Ethics Committee.

### Patient and public involvement

The COG-UK HOCI study was designed between April and May 2020 and was initially intended to run during the first wave of the COVID-19 epidemic in the UK, and, therefore, for timing and safety reasons, patients with COVID-19 were not directly included to participate in the study's design development.

### Ethics and dissemination

This study involves human participants and was approved by National Research Ethics Service Committee—Cambridge South: REC 20/EE/0118Consent for participant (both patient and healthcare worker) involvement will not be sought for COG-UK HOCI study. This approach relies on the Health Service (Control of Patient Information) Regulations 2002 (SI 1438), specifically Regulation 3 (Communicable disease and other risks to public health), and Regulation 7 (the processing of confidential information for medical research). This approach was reviewed and approved by a Research Ethics Committee.

### Post-study access to data

The terms of the funding require the COG-UK HOCI study data set to be shared on UCL's Data Repository, so that the anonymised individual participant data may be reused on an open policy by other researchers. This will be done within 6 months of public reporting of results and available for 5 years.

**Author affiliations**
[1]Comprehensive Clinical Trials Unit, University College London, London, UK
[2]Institute for Global Health, University College London, London, UK
[3]School of Psychology & Health, University of Strathclyde, Glasgow, UK
[4]Department of Infectious Disease, Imperial College London, London, UK
[5]Department of Virology, Sheffield Teaching Hospitals NHS Foundation Trust, Sheffield, UK
[6]Queen Elizabeth University Hospital, NHS Greater Glasgow and Clyde, Glasgow, UK
[7]Department of Infection, Immunity and Cardiovascular Disease, The University of Sheffield, Sheffield, UK
[8]Department of Infectious Diseases, Guy's and St Thomas' NHS Foundation Trust, London, UK
[9]Institute of Child Health, University College London, London, UK

**Acknowledgements** Our thanks to the Principal Investigators: Dr Anu Chawla (Liverpool University Hospitals NHS Foundation Trust); Dr Maria-Teresa Cutino-Moguel (Barts Health NHS Trust); Prof Alison Holmes (Imperial College Healthcare NHS Trust); Dr Nick Machin (Manchester University NHS Foundation Trust); Dr Nikunj Mahida (Nottingham University Hospitals NHS Trust); Dr Taibtha Mahungu (Royal Free NHS Foundation Trust); Dr Gaia Nebbia (Guy's and St Thomas' NHS Foundation Trust); Dr Cassie Pope (St George's University Hospitals NHS Foundation Trust); Dr Kordo Saeed (University Hospital Southampton NHS Foundation Trust); Dr Tranprit Saluja (Sandwell and West Birmingham NHS Trust); Dr Gee Yen Shin (University College London Hospitals NHS Foundation Trust); Dr Thushan de Silva (Sheffield Teaching Hospitals NHS Foundation Trust); Dr Yusri Taha (Newcastle Hospitals NHS Foundation Trust); Prof Emma Thomson (NHS Glasgow and Greater Clyde) and teams for their time and support. We also acknowledge the support of the independent members of the Joint Trial Steering Committee and Data Monitoring Committee (TSC-DMC): Prof Marion Koopmans (Erasmus MC), Prof Walter Zingg (University of Geneva), Prof Colm Bergin (Trinity College Dublin), Prof Karla Hemming (University of Birmingham), Prof Katherine Fielding (LSHTM). As well as TSC-DMC non-independent members: Prof Nick Lemoine (NIHR CRN), Prof Sharon Peacock (COG-UK). We would also thank members of COG-UK who have directly supported the study: Dr Ewan Harrison (Cambridge University), Dr Katerina Galai (PHE), Dr Francesc Coll (LSHTM), Dr Michael Chapman (HDR-UK), Prof Thomas Connor and team (Cardiff University), Prof Nick Loman and team (University of Birmingham). We also thank the COG-UK Consortium, and the UK National Institute for Health Research Clinical Research Network (NIHR CRN).

**Collaborators** The COVID-19 Genomics UK (COG-UK) Consortium: MRC–University of Glasgow Centre for Virus Research (CVR), Glasgow, UK: Dr Joshua Singer, Dr Joseph Hughes and Prof Emma Thomson in supporting design and implementation of the HOCI sequence reporting tool. University of Sheffield, Sheffield, UK: Dr Matthew Parker supporting design and testing of the HOCI sequence reporting tool. UCL, London, UK: Dr Asif Tamuri, Dr Sunando Roy and Dr Stefan Piatek supporting design, implementation and testing of the HOCI sequence reporting tool.

**Contributors** JB developed the study protocol and contributed to the writing of the manuscript. OS and AC were involved in development of the statistical analysis facets of the trial and contributed to the writing of the manuscript. MP was involved in development of the health economic analysis facets of the study and contributed to the writing of the manuscript. PF and FM were involved in development of the process evaluation of the study and contributed to the writing of the manuscript. LH is the trial manager for the study. JP, DP, CP, TdS, GN and LBS are responsible for the operationalisation of study procedures. RM will contribute to data collection and management. JBr is the chief investigator for the study. All authors critically reviewed and approved the final version.

**Funding** The COG-UK HOCI study is funded by the COG-UK Consortium, which is supported by funding from the Medical Research Council (MRC) part of UK Research & Innovation (UKRI), the National Institute of Health Research (NIHR) and Genome Research Limited, operating as the Wellcome Sanger Institute (UKRI MRC Grant number MC PC 19027).

**Competing interests** None declared.

**Patient and public involvement** Patients and/or the public were not involved in the design, or conduct, or reporting, or dissemination plans of this research.

**Patient consent for publication** Not applicable.

**Provenance and peer review** Not commissioned; externally peer reviewed.

**ORCID iDs**
James Blackstone http://orcid.org/0000-0003-4335-5269
Fiona Mapp http://orcid.org/0000-0003-0733-6036
Andrew Copas http://orcid.org/0000-0001-8968-5963
Paul Flowers http://orcid.org/0000-0001-6239-5616
Leanne Hockey http://orcid.org/0000-0001-5716-7643
David Partridge http://orcid.org/0000-0002-0417-2016
Thushan de Silva http://orcid.org/0000-0002-6498-9212
Gaia Nebbia http://orcid.org/0000-0002-7524-1947
Luke B Snell http://orcid.org/0000-0002-6263-9497

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
