## [Reviewer comments · BMJ Open]

ARTICLE DETAILS

TITLE (PROVISIONAL)	Protocol for the COG-UK hospital onset COVID-19 infection (HOCI) multicentre interventional clinical study: evaluating the efficacy of rapid genome sequencing of SARS-CoV-2 in limiting the spread of COVID-19 in United Kingdom NHS hospitals
AUTHORS	Blackstone, James; Stirrup, Oliver; Mapp, Fiona; Panca, Monica; Copas, Andrew; Flowers, Paul; Hockey, Leanne; Price, James; Partridge, David; Peters, Christine; de Silva, Thushan; Nebbia, Gaia; Snell, Luke; McComish, Rachel; Breuer, Judith

VERSION 1 – REVIEW

REVIEWER	Charles, Myrna Centers for Disease Control and Prevention, NCIRD/Influenza Division
REVIEW RETURNED	20-Jul-2021

GENERAL COMMENTS	How will HCWs be identified and enrolled at these sites? Is there routine testing of HCWs (at the NHS sites) to know their SARS-CoV-2 status?
---

REVIEWER	Almuedo, Alex ISGlobal, International Health
REVIEW RETURNED	18-Aug-2021

GENERAL COMMENTS	This manuscript exposes an interesting and valuable multicentre prospective interventional protocol that aims to evaluate if viral sequencing informed by two proposed interventions could contribute to clarify the origin of SARS-CoV-2 infection (communitarian versus hospital acquired). The hypothesis and objective of the study is really of great interest and can improve nosocomial transmissions management of SARS-Cov-2 infection In general terms all the protocol is clear and well described. The SPIRIT Checklist it is well and correctly addressed. Background section is correct however some articles about genome sequencing and nosocomial infection have been recently published. It is clear that the protocol and manuscript probably were written before the publication and that is the reason why it lacks of reference. If that is the reason I would highly recommend to add the date of protocol 5.0 version in line 53 of Page2. Another strategy could be to refer at least one the recently published. Although during the protocol it is mentioned that possible changes in infection prevention and control (IPC) practices can be observed due to genome sequencing it is not clear how these changes will be informed or measured during the analysis. Being part of the
---

	hypothesis of the study and also part of secondary objectives, it would be desirable to because it is also one of aims of the study. It would be recommendable to add also in order to analyse possible results differences between heterogenous sites. It would be desirable to add some information of how the samples will be collected in all sites. Strengths and limitations are well addressed. The rest of the section of the manuscripts are clear. It is an stimulating protocol and I looking forward to see the results.
--	---

VERSION 1 – AUTHOR RESPONSE

Reviewer 1

Reviewer question: How will HCWs be identified and enrolled at these sites? Is there routine testing of HCWs (at the NHS sites) to know their SARS-CoV-2 status?

Author response: HCWs will not be enrolled as index cases for the HOCl trial due to the medico-legal and logistical challenges involved in their inclusion. Routine testing of HCWs has been varied across NHS hospitals throughout the Covid-19 pandemic, with some sites routinely making their HCW data available for use in the reference sequence data. This has been clarified in Population section.

Reviewer 2

Reviewer comment: This manuscript exposes an interesting and valuable multicentre prospective interventional protocol that aims to evaluate if viral sequencing informed by two proposed interventions could contribute to clarify the origin of SARS-CoV-2 infection (communitarian versus hospital acquired). The hypothesis and objective of the study is really of great interest and can improve nosocomial transmissions management of SARS-Cov-2 infection. In general terms all the protocol is clear and well described. The SPIRIT Checklist it is well and correctly addressed.

Author response: Thank you; it is encouraging to hear that the work proposed here is considered salient and useful.

Reviewer comment: Background section is correct however some articles about genome sequencing and nosocomial infection have been recently have been published. It is clear that the protocol and manuscript probably were written before the publication and that is the reason why it lacks of reference. If that is the reason I would highly recommend to add the date of protocol 5.0 version in line 53 of Page2. Another strategy could be to refer at least one the recently published.

Author response: Yes, the protocol paper submission was made in April-2021. However, we have now updated references in the Background section regarding the literature on nosocomial infection and preventative genomic sequencing efforts for SARS-CoV-2. We have deleted the statement “To date, all studies have been retrospective”, and added “Although some studies have described the prospective use of viral sequencing to inform infection control for SARS-CoV-2 [Meredith et al. Moore et al., Page et al.], none have prospectively evaluated the impact of sequencing on the incidence of nosocomial infection or on infection control actions across all cases with hospital onset”. We have also added a citation to a review article by Abbas et al. in the opening paragraph. The protocol version on page 2 has also been updated to the most recent approved iteration (v6.0) with date added.

Reviewer comment: Although during the protocol it is mentioned that possible changes in infection prevention and control (IPC) practices can be observed due to genome sequencing it is not clear how these changes will be informed or measured during the analysis. Being part of the hypothesis of the study and also part of secondary objectives, it would be desirable to because it is also one of aims of the study. It would be recommendable to add also in order to analyse possible results differences between heterogenous sites.

Author response: Given we have the benefit of prospective data collection, a set of specific Case Report Forms (CRFs) have been designed to allow the collection of granular data surrounding IPC practices, which will then be compared between phases in the analysis. This has been clarified in the Outcomes section.

Reviewer question: It would be desirable to add some information of how the samples will be collected in all sites.

Author response: Patient samples are collected and sequenced as part of pre-existing arrangements between NHS Trusts and COG-UK to support national genomic surveillance efforts. This has now been included in the Recruitment section.

Reviewer comment: Strengths and limitations are well addressed. The rest of the section of the manuscripts are clear. It is an stimulating protocol and I looking forward to see the results.

Author response: Thank you.

VERSION 2 – REVIEW

REVIEWER	Almuedo, Alex ISGlobal, International Health
REVIEW RETURNED	17-Mar-2022

GENERAL COMMENTS	This manuscript exposes an interesting and valuable multicentre prospective interventional protocol that aims to evaluate if viral sequencing informed by two proposed interventions could contribute to clarifying the origin of SARS-CoV-2 infection (communitarian versus hospital-acquired). The hypothesis and objective of the study is really of great interest and can improve nosocomial transmissions management of SARS-Cov-2 infection In general terms, all the protocol is clear and well described. The SPIRIT Checklist is well and correctly addressed. The review of the new version of the manuscript shows that the authors carefully address all the comments from Editors and reviewers. I can not see and explore Figure 1. Although it can be a problem with my computer I highly recommend reviewing this fact. Otherwise, I think the manuscript is suitable for publication.
---